# INVERSESCOPE: SCALABLE ACTIVATION INVERSION FOR INTERPRETING LARGE LANGUAGE MODELS

## ABSTRACT

Understanding the internal representations of large language models (LLMs) is a central challenge in interpretability research. Existing feature interpretability methods often rely on strong assumptions about the structure of representations that may not hold in practice. In this work, we introduce InverseScope, an assumption-light and scalable framework for interpreting neural activations via input inversion. Given a target activation, we define a distribution over inputs that generate similar activations and analyze this distribution to infer the encoded information. To address the inefficiency of sampling in high-dimensional spaces, we propose a novel conditional generation architecture that significantly improves sample efficiency compared to previous method. We further introduce a quantitative evaluation protocol that tests interpretability hypotheses using the feature consistency rate computed over the sampled inputs. InverseScope scales inversion-based interpretability methods to larger models and practical tasks, enabling systematic and quantitative analysis of internal representations in real-world LLMs.

## 1 INTRODUCTION

Recent advances in mechanistic interpretability aim to reverse-engineer neural networks' computations into human-understandable processes (Bereska & Gavves, 2024; Sharkey et al., 2025). A central task in this field is feature interpretability, which seeks to understand what information is encoded in a network's activations and how it is represented. This understanding is crucial for analyzing how information propagates and is processed across layers. Numerous methods have been proposed for feature interpretability, including linear probing (Alain & Bengio, 2016; Park et al., 2023), sparse dictionary learning (Cunningham et al., 2023; Gao et al., 2024), and other approaches (Bau et al., 2017; Xu et al., 2024).

Despite their successes, these methods share a fundamental limitation: they rely on strong assumptions about the structure of neural representations. Specifically, linear probing assumes a linear relationship between activations and specific concepts in the inputs, and sparse dictionary learning presupposes that activations can be decomposed into a sparse sum of linear directions. The validity of these assumptions remains an open and actively debated question, particularly in the context of LLMs (Levy & Geva, 2024; Engels et al.; Smith, 2024). Designing experiments that rigorously test these hypotheses is itself a challenging problem, making it difficult to assess the reliability of interpretations derived from such approaches.

These limitations highlight the need for interpretability methods that rely on minimal assumptions about the structure of neural representations. One promising direction is to invert activation back to the input space, where human intuitions are more naturally grounded. The strategy of connecting activations back to the inputs that produce them has a long-standing history in interpretability research, including early work on activation maximization (Erhan et al., 2009; Nguyen et al., 2016) and neural representation inversion (Mahendran & Vedaldi, 2014). Building on this line, InversionView (Huang et al., 2024b) adapts the idea of previous inversion-based methods to language models by interpreting the information encoded in an activation through the collection of inputs that generates similar representations. These methods enable feature interpretability by relying only on the geometric proximity of activations, without presupposing restrictive structural assumptions like linearity or sparsity.

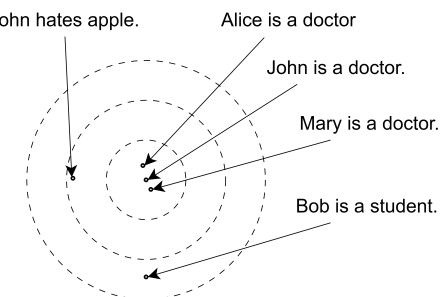

Figure 1: An toy example of samples in activation space and their corresponding inputs.

While prior works have demonstrated the feasibility of interpreting activations through distributions of matching inputs, such efforts have largely been limited to small-scale models and qualitative case studies, lacking scalability to real-world settings. This motivates our work to advance inversion-based interpretability by scaling it to larger open-source LLMs and applying it to practical tasks.

The main contributions of this paper are as follows:

- We introduce InverseScope, a novel conditional generation architecture that substantially improves sampling efficiency for input sampling, making inversion-based methods practical for LLMs with up to 27B-parameters.
- We establish a rigorous, quantitative evaluation framework that systematically assesses the information revealed in the input distributions recovered via inversion.
- We apply our framework to in-context learning tasks, revealing new mechanistic insights into the generation and disappearance of task-level features.

Collectively, our work substantially expands the reach of inversion-based interpretability, scaling this powerful, assumption-light approach to 27B-parameter LLMs and complex in-context learning tasks. This enables the field to move beyond illustrative examples toward a more systematic and rigorous analysis of feature representations.

## 2 METHOD DESCRIPTION

Our method builds on a simple observation: similar activations tend to encode semantically similar information (Bengio et al., 2013). In this paper, we use the term "activation" to specifically denote the output of a network layer. If two distinct inputs produce nearly identical activations, then the differences between those inputs are unlikely to be distinguishable or encoded in that representation. Conversely, if nearby activations consistently correspond to inputs sharing a particular feature, this indicates that the feature is encoded within that region of latent space. This reasoning is fundamentally grounded in the continuity of neural networks: since downstream layers apply continuous transformations to their inputs, activations that are close in space are functionally equivalent from the network's perspective.

Building on this observation, our method investigates the information encoded in a target activation $\hat{z}$ by inverting the activation geometry to recover the distribution of functionally equivalent inputs. By inspecting this distribution, we can generate hypotheses about which features might be encoded in that activation. Once a hypothesis is proposed, we employ quantitative metrics to evaluate whether the feature is consistently reflected in the distribution, thereby rigorously testing the feature's representation in the latent space.

For example, as illustrated in Figure 1, consider an input–activation pair $(\hat{x}, \hat{z})$, where $\hat{x}$ is "John is a doctor." and $\hat{z}$ is the activation it produces. Our method assigns higher weights to inputs such as "Mary is a doctor." and "Alice is a doctor.", which yield activations close to $\hat{z}$, while downweighting inputs like "John hates apples." that produce activations farther away. Analyzing this reweighted distribution allows us to hypothesize that $\hat{z}$ encodes features like "Subject is a doctor" or, more generally, "Subject's profession." Then, we can formalize this feature and use our quantitative framework to rigorously test its consistency across the latent space.

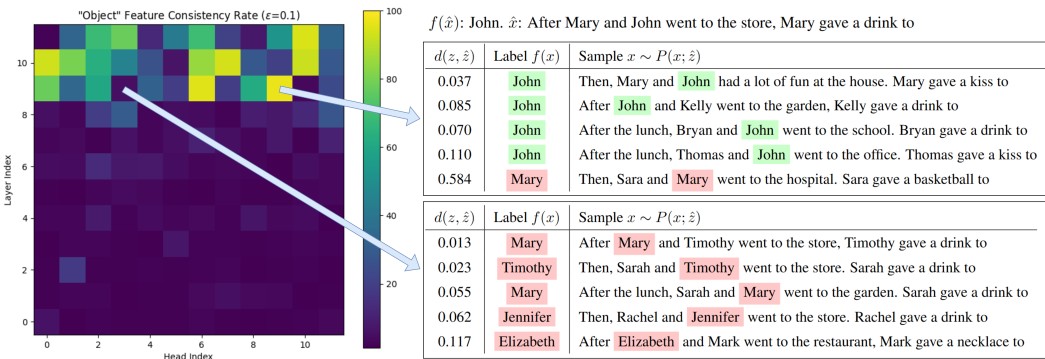

Figure 2: Example results from applying our method to the indirect object identification task on GPT-2-small. **Left:** Feature consistency rates for the object feature, computed using $\hat{z}$ extracted from the outputs of different attention heads across layers. **Right:** Example samples from the conditional distribution $P(x; \hat{z})$ for specific activations $\hat{z}$ from heads L9H3 and L9H9, annotated with their distance $d(z(x), \hat{z})$ and the corresponding feature label $f(x)$.

Formally, the reweighting procedure is defined by a probability distribution $P(x; \hat{z})$ over all possible inputs, assigning higher probability to inputs whose activations $z(x)$ lie closer to a reference activation $\hat{z}$ under a chosen distance metric. Mathematically, given a target activation $\hat{z} \in \mathbb{R}^n$, we define the probability of sampling an input $x$ by:

$$P(x; \hat{z}) \propto k(d(z(x), \hat{z})/\epsilon).$$

where $d$ is a metric over activation space $\mathbb{R}^n$, and $k$ is a kernel function (e.g., a Gaussian $k(d) = \exp(-d^2)$ or a hard threshold $k(d) = \mathbb{I}_{\{d<1\}}$, as used in Huang et al. (2024b)), and $\epsilon$ is a bandwidth parameter controlling the neighborhood size.

We propose a three-step interpretation pipeline based on the defined distribution $P(x; \hat{z})$, which consists of hypothesis generation, formalization, and evaluation:

1. (Optional) **Sample inputs from $P(x; \hat{z})$.** Human analysts or auxiliary models (e.g., LLMs) can summarize the commonalities in these samples to form hypotheses about which interpretable feature is encoded in $\hat{z}$.

2. **Define a candidate feature function.** Translate the hypothesis into a formal feature function $f : x \to f(x)$, which maps each input $x$ to a discrete, interpretable label $f(x)$. This function operationalizes the interpretable concept we aim to test.

3. **Evaluate $f$ via feature consistency rate.** We measure how consistently the feature $f$ is represented in the local activation neighborhood by computing the feature consistency rate (FCR):

$$\text{FCR}(f) = \mathbb{E}_{(\hat{x}, \hat{z}) \sim \mathcal{D}} \mathbb{E}_{x \sim P(x; \hat{z})} \mathbb{I}_{\{f(x) = f(\hat{x})\}}. \tag{1}$$

where $\mathcal{D}$ denotes a predefined dataset distribution over inputs and their corresponding activations. The FCR is the expected probability that a sampled input $x$ from the neighborhood of $\hat{z}$ possesses the same feature $f$ as the seed input $\hat{x}$. A high feature consistency rate indicates that the feature $f$ is consistently preserved across local neighborhoods in activation space, suggesting it is reliably encoded in the activations.

For scenarios where the feature of interest is known in advance, steps 2 and 3 can be applied directly to quantitatively assess whether the target feature is represented in the given activations. We recognize that step 2 – formalizing a feature into a function – may appear abstract in general terms. We provide a more detailed explanation in Appendix A.

Figure 2 illustrates an example of the results obtained with our method on the indirect object identification task. In this case, we analyze the "object feature," formalized as a function $f$ that maps the input $x$ to the object's name. We compute the feature consistency rate (Equation 1) for activations extracted from different attention heads. The results reveal a notable variation in feature consistency

across attention heads, indicating that certain sites encode the object feature significantly more reliably than others. A more detailed discussion of these results is provided in Section 4.1.

The key challenge in implementing this evaluation lies in efficiently sampling from $P(x; \hat{z})$. In the high-dimensional activation spaces typical of modern LLMs, the probability that a random input produces an activation close to $\hat{z}$ decays exponentially with dimensionality. This renders naive rejection sampling prohibitively inefficient for practical applications, particularly with LLMs where activation dimensions can reach thousands.

To address this fundamental bottleneck, we introduce InverseScope, a novel conditional generation architecture. InverseScope efficiently generates inputs guaranteed to produce activations close to the target $\hat{z}$, effectively overcoming the sampling bottleneck that plagues naive inversion methods.

## 3 INVERSESCOPE

In this section we describe **InverseScope**, including its network architecture and training procedure. Our objective is to efficiently sample from the conditional distribution $P(x; \hat{z})$ defined in the previous section. To achieve this, InverseScope is designed as a conditional generator trained to approximate this target distribution.

We adopt the decoder-only Transformer paradigm, which has proven highly effective for modeling natural language distributions via next-token prediction objectives (Brown et al., 2020). InverseScope extends this standard language modeling framework by conditioning the token prediction not only on the preceding sequence context but critically, also on the target activation $\hat{z}$. This conditioning is essential to align the generated sequence with the semantic information encoded in $\hat{z}$, allowing the generator to approximate the target distribution $P(x; \hat{z})$.

### 3.1 NETWORK ARCHITECTURE

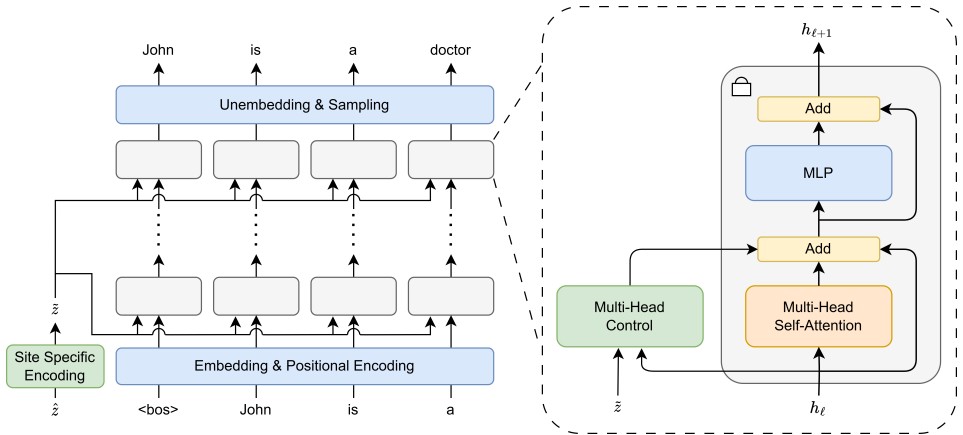

Figure 3: Network architecture of InverseScope. The decoder-only Transformer backbone is shown with its parameters colored in orange and blue. The additional control and site-encoding layers introduced for conditioning are colored in green.

As illustrated in Figure 3, our conditional generator is based on a standard decoder-only Transformer, augmented with additional Control Layers that transmit the conditioning information from activation $\hat{z}$ into the model's hidden states. These Control Layers operate analogously to the cross-attention mechanism found in encoder-decoder Transformers.

The mathematical formulation of the control layers is as follows:

$$q_\ell^{(i)} = Q_\ell h_\ell^{(i)} + \bar{q}_\ell, \ k_\ell = K_\ell \hat{z} + \bar{k}_\ell, \ v_\ell = V_\ell \hat{z} + \bar{v}_\ell,$$

$$\omega_\ell^{(i)} = \tanh(\langle q_\ell^{(i)}, k_\ell \rangle),$$

$$\text{Control}_\ell(h_\ell^{(i)}, \hat{z}) = \omega_\ell^{(i)} v_\ell,$$

where $i$ is the inference position index. The parameter matrices $Q_\ell, K_\ell, V_\ell$ are distinct from the self-attention parameters of the backbone model, and $\bar{q}_\ell, \bar{k}_\ell, \bar{v}_\ell$ are the corresponding bias terms. For simplicity, we omit the layer norm and only provide formulation for the single-head version. In practice, we use a multi-head variant where each head has its own query, key, value matrices and their outputs are summed to produce the final control signal. This signal is then added to the hidden state $h_\ell^{(i)}$, after the standard self-attention operation.

**Condition on different sites.** In practice, we are interested in interpreting activations collected from multiple sites in the target model. By site, we mean a specific location in the network, such as the output of the $\ell$-th decoder layer at the last inference position. Training a separate conditional generator for each site would be prohibitively expensive. Instead, we design a shared conditional generator that supports conditioning on activations from arbitrary sites.

To achieve this, we introduce a series of site-specific projection layers, as shown in the left-bottom corner of Figure 3. Each activation $\hat{z}$ is first pass through a learned, site-specific linear transformation before being sent to the shared generator. These linear layers project the conditions from different sites into a common latent space, allowing the core generator to operate uniformly regardless of the origin of $\hat{z}$.

By combining novel Control Layers with parameter sharing for handling activations from arbitrary sites, InverseScope provides a flexible and scalable architecture for conditional generation. As demonstrated in Section 4.1, this design enables us to more accurately approximate the complex, high-dimensional distributions $P(x; \hat{z})$, leading to lower refusal rates and more efficient sampling. This capability is essential for scaling inversion-based interpretability methods to large models and diverse activation sites.

### 3.2 DATASET AND TRAINING

In the applications considered in this paper, we focus on task specific input distributions, such as all possible inputs for the indirect object identification task, or all possible in-context learning prompts for translation tasks, rather than the general distribution of all possible natural languages. We use $\mathcal{P}(\mathcal{X})$ to denote this task-specific prior input distribution.

Given a target model and a task-specific prior $\mathcal{P}(\mathcal{X})$, we construct a training dataset by collecting input–activation pairs $(\hat{x}, \hat{z})$, where $\hat{x} \sim \mathcal{P}(\mathcal{X})$ and $\hat{z} = z(\hat{x})$ denotes the activation at a specified site within the model. To prevent the conditional generator from collapsing to a degenerate solution and just fitting the delta function $\delta(\hat{x})$, we inject noise into the collected activations. As a result, the final training dataset takes the form $\{(\hat{x}_i, \hat{z}_i + r_i)\}_{i=1}^N$. Ideally, the injected noise $r_i$ should be sampled to match the kernel function used to define $P(x; \hat{z})$. A more detailed discussion of this procedure is provided in Appendix B.

For each task, we perform a 2-step training. First fine-tune the backbone, then train the additional layers independently. In the first step, we fine-tune a decoder-only Transformer on $\mathcal{P}(\mathcal{X})$, which will serve as the backbone of our conditional generator (corresponding to the non-green layers in Figure 3). This step follows standard supervised fine-tuning procedures. Across all experiments in this paper, we use GPT-2-small (Radford et al., 2019) as our backbone model, regardless of which target model is being interpreted.

During the subsequent training phase, we freeze the backbone parameters and train only the additional layers introduced in Section 3.1 (corresponding to the green layers in Figure 3). By decoupling the supervised fine-tuning of the backbone from the training of the conditional layers, we ensure that the control layers are dedicated to capturing distinctions between different conditioning activations $\hat{z}$. More detailed training settings are provided in Appendix C.

The training objective for the control layers is to maximize the conditional log-likelihood of the training input $\hat{x}$ given the noisy activation:

$$\max_\theta \frac{1}{N} \sum_i \log P_\theta(\hat{x}_i; \hat{z}_i + r_i),$$

where $\theta$ denote the parameters of the control layers. This objective can be decomposed into standard next-token prediction loss.

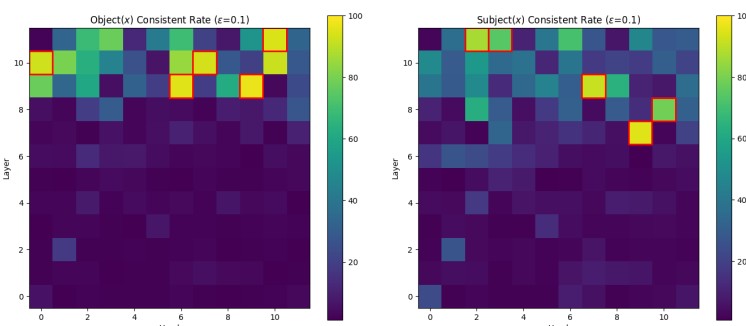

Figure 4: Feature consistency rate of the outputs of GPT-2-small's attention heads in the IOI task. **Left:** consistency rate of object feature. **Right:** consistency rate of subject feature. The top-5 attention heads with the highest feature consistency rates are marked with red rectangular.

## 4 EXPERIMENTS

In this section, we evaluate our method across a range of tasks and target models, benchmarking its performance against an inversion method and a sparse dictionary learning method. We begin with the indirect object identification (IOI) task (Wang et al., 2022) as a case study, showcasing InverseScope's significant gains in sampling efficiency compared to the inversion baseline. Next, we benchmark our approach against sparse dictionary learning on the RAVEL dataset (Huang et al., 2024a), demonstrating that InverseScope yields a clearer signal of the information encoded in activations. Finally, we apply our method to in-context learning tasks (Hendel et al., 2023), providing a mechanistic explanation for a phenomenon previously observed but not understood.

### 4.1 CASE STUDY: INDIRECT OBJECT IDENTIFICATION

In this subsection, we apply our method to the indirect object identification (IOI) task (Wang et al., 2022) on GPT-2-small (Radford et al., 2019). IOI serves as a case study for two reasons. First, the mechanism underlying GPT-2-small's behavior on IOI has been carefully dissected in prior work (Wang et al., 2022; Makelov et al., 2024), providing a strong reference point against which to validate our findings. Second, it offers a standard benchmark for evaluating sampling efficiency, allowing for a direct comparison with the inversion-based baseline of Huang et al. (2024b). A more detailed experimental setting is provided in Appendix D.1.

For IOI, the input distribution $\mathcal{P}(\mathcal{X})$ is defined as a uniform distribution over template-generated sentences such as $\hat{x}$ = "When [A] and [B] went to the store, [A] gave a drink to". We analyze attention head outputs across layers at the final inference position—i.e., the position that takes the token "to" as input. This yields a total of 144 activation sites in GPT-2-small (12 layers with 12 heads per layer). Our objective is to characterize the information encoded in these activations and to identify which attention heads contribute features that enable GPT-2-small to correctly resolve the indirect object and generate the appropriate name.

To solve IOI, two features are naturally hypothesized: the subject feature, $\mathrm{Subject}(x)$, which maps $x$ to the repeated name, and the object feature, $\mathrm{Object}(x)$, which maps $x$ to the correct indirect object to be predicted. In this case, Step 2 of our method is straightforward, as both features can be formalized using simple rule-based functions. Step 3 is then carried out by evaluating feature consistency to identify which attention heads aggregate information about these features into the final inference position.

As shown in Figure 4, the object feature is primarily encoded by attention heads in layers 9 to 11. The top 5 heads with the highest feature consistency rates are L9H6, L9H9, L10H0, L10H7, and L11H10—precisely the Name Mover Heads and Negative Name Mover Heads identified in Wang et al. (2022). Other heads with high feature consistency rates also coincide with the Backup Name Mover Heads. In contrast, the subject feature is encoded earlier in the network, such as L7H9 and L8H10, which correspond to the Subject Inhibition heads also described in Wang et al. (2022).

These results support the circuit discovered in Wang et al. (2022), that the model first aggregates the subject feature to the last position, then is the object feature.

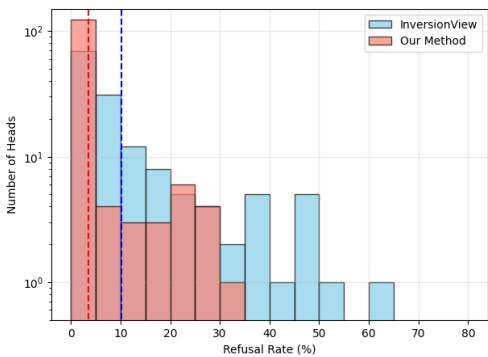

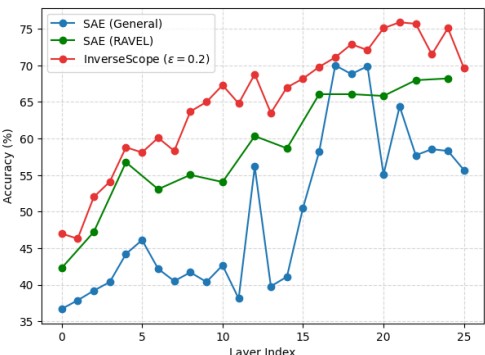

Figure 5: Histogram of sample refusal rates across GPT-2-small's 144 attention heads in the IOI task. The vertical dashed line indicates the average refusal rate across all heads. Lower refusal rates correspond to higher sample efficiency. The y-axis is log-scaled for illustrative purpose.

Figure 6: The attribute classification accuracy of SAE (General), SAE (RAVEL) and InverseScope for residual stream activations across different layers. InverseScope achieves consistently higher accuracy than both SAE baselines across all layers.

**Comparison with other inversion-based method.** We compare the sample efficiency of our method against InversionView (Huang et al., 2024b). To ensure a fair comparison, we follow the same experimental setup as Huang et al. (2024b), using the kernel function $k(d) = \mathbb{I}_{\{d < \epsilon\}}$. Under this setting, $P(x; \hat{z})$ defines a uniform distribution over all inputs whose activations $z(x)$ lie within the $\epsilon$−neighborhood $B_\epsilon(\hat{z})$. We evaluate sample efficiency by computing the refusal rate—the proportion of samples provided by different method's conditional generator for which $z(x) \notin B_\epsilon(\hat{z})$. A lower refusal rate corresponds to higher sampling efficiency.

As shown in Figure 5, our method consistently improves sample efficiency across all attention heads. We reduce the average refusal rate from 10.2% to 3.5%. In the worst case, we significantly reduce the refusal rate from 60.5% to 31.7%. This demonstrates the robustness of our approach. Such robustness is especially important for scaling to larger models and more complex tasks, where high-refusal-rate cases are more likely to arise. Ensuring that the generator can approximate more complex $P(x; \hat{z})$ is critical for making inversion-based methods viable at scale.

Beyond the analyses presented above, we provide additional experimental results in Appendix E.1. These include extended qualitative examples of sampled inputs from representative attention heads, further visualizations illustrating how feature consistency evolves with the activation distance, and an ablation study examining the effect of the kernel width parameter $\epsilon$. These supplementary results offer a more comprehensive view of the activation neighborhoods and complement the main findings reported in this section.

## 4.2 ATTRIBUTE IDENTIFICATION EVALUATIONS

In this subsection, we evaluate our method's efficacy in attribute identification from activations, and benchmark it against the sparse autoencoder (SAE) approach (Bricken et al., 2023). SAEs are unsupervised models trained to reconstruct activations under a sparsity constraint, with the goal of decomposing them into monosemantic, interpretable features. We focus on residual stream activations of Gemma-2-2B (Team et al., 2024) on the RAVEL dataset (Huang et al., 2024a), a dataset specifically designed to assess attribute identification fidelity. We demonstrate that our method provides a clearer signal of the encoded attributes compared with SAEs.

The SAE baselines we compare against are: (1) a series of 16k-wide SAEs from gemma-scope-2b-res (Lieberum et al., 2024), referred to as **SAE (General)**; and (2) a series of 4k-wide JumpReLU SAEs trained specifically on the RAVEL training set, referred to as **SAE (RAVEL)**. We include the

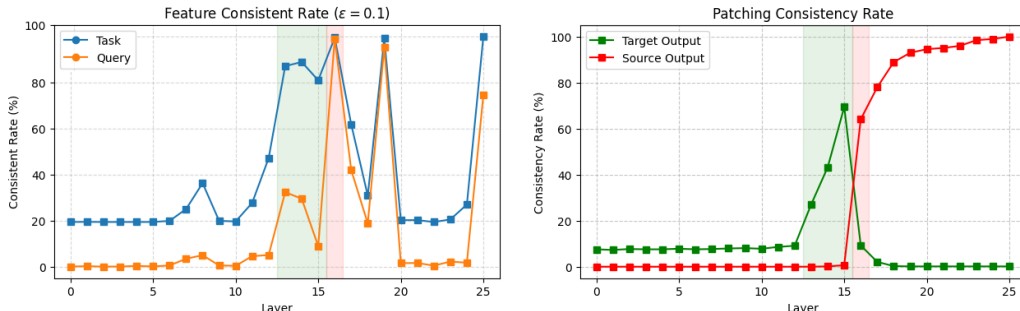

Figure 7: **Left:** Feature consistency rates of the outputs of Gemma-2-2B's attention layers in the ICL task. Blue represents the task feature. Orange represents the query feature. **Right:** Results of task-vector patching experiments. Green represents the rate at which the patched inference produces the correct target output, which indicate the activation encoded the abstract task. Red represents the rate at which it incorrectly produces the source output, which indicate the activation encoded the specific output token.

latter for a fair comparison, since the InverseScope model evaluated here is also trained only on the RAVEL training dataset. Additional experimental details are provided in Appendix D.2.

RAVEL provides prompts designed to elicit concept-related attributes (e.g., "language" or "continent" for a given city). For instance, the prompt "People in [City] usually speak" is used to probe the "language" attribute of the "city" entities. Crucially, while the original benchmark evaluates interpretability via causal interventions on model behavior, we instead focus on a more fundamental question: assessing the method's fidelity in identifying the correct attribute encoded within the activation itself. Specifically, we target residual stream activations, focusing on the last token position across all decoder layers.

We frame this benchmark as a classification task. For SAE, this involves identifying a representative feature for each attribute, which then serves as a simple classifier: when the feature activates, it indicates the presence of the corresponding attribute. In our method, we classify an attribute based on the attributes of similar inputs generated by InverseScope. Given a target activation $\hat{z}$, we sample a candidate input $x \sim P(x; \hat{z})$ with InverseScope. If $x$ exhibits a recognizable attribute, we classify $\hat{z}$ as encoding that attribute; otherwise, we assign a null attribute, counting it as a failed classification. Notably, the classification accuracy in this setup corresponds exactly to the feature consistency rate computed in Step 3 of our pipeline. Additional experimental details for SAE baselines and InverseScope classification are provided in Appendix D.2.

As shown in Figure 6, InverseScope achieves consistently higher accuracy than both SAE baselines across all layers. This indicates that InverseScope can recover attribute information that SAEs fail to capture. Moreover, both InverseScope and SAE (RAVEL) reveal a gradual accumulation of attribute information across layers, whereas SAE (General) suggests more abrupt shifts. Given the residual connection structure of the model, we believe the smoother progression observed with InverseScope better reflects the underlying mechanisms of attribute representation.

Additional results are provided in Appendix E.3, where we extend our evaluation to Gemma-2 models of larger sizes, including 9B and 27B. These experiments examine how InverseScope scales with hidden dimensionality and include further visualizations and analyses. Together, these supplemental results demonstrate that InverseScope remains sample-efficient and reliable when applied to realistic, open-source LLMs.

### 4.3 UNDERSTANDING IN-CONTEXT LEARNING

In this subsection, we apply our method to investigate the mechanism of in-context learning (ICL), aiming to explain a phenomenon discovered prior interpretability research (Hendel et al., 2023). We conduct experiments on Gemma-2-2B (Team et al., 2024) and LLaMA-2-7B (Touvron et al., 2023) using a synthetic ICL translation dataset. The main results for Gemma-2-2B are presented here,

while the results for LLaMA-2-7B are deferred to Appendix D.3, where we also provide further details of the experimental setup.

ICL is a well-known emergent capability of LLMs, in which the model generalizes from a few input–output examples presented in the prompt to perform the same task on a new input. Prior work has shown that residual stream activations at intermediate layers can encode abstract task-level representations, referred to as task vectors (Hendel et al., 2023).

However, as shown by the green curve in Figure 7 (right), only a narrow range of layers exhibit this property. In shallower layers, residual activations do not appear to encode task-related features, while in deeper layers the residual activations instead capture specific output tokens rather than the abstract ICL task, as illustrated by the red curve in Figure 7 (right). Due to space limitations, a detailed description of the original task-vector experiment is provided in Appendix D.3.

This raises a natural question: why do task vectors emerge specifically at these specific layers? While Hendel et al. (2023) identifies the phenomenon, it does not provide a grounded explanation. We will demonstrate that our method can help to explain such phenomenon, by analyzing the information encoded in the output activations of attention layers.

Similar to the IOI task, we analyze the output activations of attention layers to identify where important features of the ICL task are aggregated. The first hypothesized feature is the task feature $\mathrm{Task}(x)$, which maps $x$ to the task demonstrated in the prompt (e.g., $\mathrm{Task}(x) =$ "English-to-French translation"). The second is the query feature $\mathrm{Query}(x)$, which maps $x$ to the specific query that needs to be processed (e.g., the English word to be translated). Since both features can be defined as rule-based functions, we can automatically compute their feature consistence rates.

The consistence rates are shown in Figure 7 (left). As the figure indicates, neither feature is consistently encoded in the outputs of the first 13 layers. Around layer 13, the task feature becomes clearly detectable, while the query feature remains largely absent. This separation continues until layer 16, at which point the query feature begins to emerge strongly.

We propose that the separation in the emergence of task and query features underlies the task vector phenomenon. In layers 13–15 (shaded green in Figure 7), attention outputs inject the task feature into the residual stream without yet incorporating the query feature. As a result, the residual stream encodes only an abstract representation of the task. After layer 16 (shaded red), the query feature accumulates, transforming the residual stream into a representation of the specific output token rather than the abstract task. The task vector does not disappear after layer 16—it is simply masked by the presence of the query feature. A similar pattern is observed in LLaMA-2-7B, as shown in Figure 11 in Appendix E.4.

## 5 RELATED WORKS

**Feature interpretability.** A variety of methods have been developed to interpret neural network features. Classical approaches such as linear probing train simple classifiers on activations to identify linearly encoded features (Alain & Bengio, 2016; Park et al., 2023). More recent work includes sparse dictionary learning, which decomposes activations into sparse and interpretable components to disentangle feature representations (Cunningham et al., 2023; Gao et al., 2024), and methods that analyze the activation by mapping them to the vocabulary space (Geva et al., 2022).

**Inversion-based interpretability.** A complementary line of research focuses on interpreting model activations by identifying the inputs that give rise to them. This approach has its roots in early work on activation maximization and representation inversion (Erhan et al., 2009; Nguyen et al., 2016; Mahendran & Vedaldi, 2014), originally developed in the vision domain. Recent efforts have extended these techniques to language models. Recent efforts such as InversionView (Huang et al., 2024b) have extended these techniques to language models.

**Nautral language interpretability.** Several recent works have explored assigning human-interpretable labels—such as natural language descriptions—to the internal activations of LLMs. Training free methods like SelfIE (Chen et al., 2024) and PatchScope (Ghandeharioun et al., 2024) use a pretrained LLM to read out the information encoded in residual stream activations. Simi-

larly, LatentQA (Pan et al., 2024) use supervise training to get a decoder model that answers natural language questions about these activations.

**Conditional and Controllable Generation.** Conditional generation, which generate outputs based on a constrained intermediate state, is a foundational technique in deep learning. Initially established with Conditional VAEs (Sohn et al., 2015) and Conditional GANs (Mirza & Osindero, 2014), this paradigm has evolved to methods where the conditioning signal acts as an explicit request for specific output structure. Noteworthy examples include modern text-to-image synthesis (Ramesh et al., 2022) and structured guidance methods like ControlNet (Zhang et al., 2023), which condition large generative models on input features. Similarly, controllable text generation (Keskar et al., 2019) steers language model output by conditioning on explicit attributes. Our work, InverseScope, extends this established conditional generation framework by repurposing the core task—mapping a constrained latent state to an output—to specifically address the challenge of scalable mechanistic interpretability via activation inversion.

## 6 LIMITATIONS

While our results demonstrate the effectiveness of InverseScope, several limitations remain. First, the method does not yet scale to long input sequences. As input length increases, the corresponding input distribution becomes substantially more complex, and our approach currently performs reliably only on inputs spanning tens of tokens. Second, although we provide a quantitative—and thus automatable—framework for evaluating feature hypotheses, generating these hypotheses still requires human involvement. Automating or systematizing this step remains an open challenge. We leave addressing these issues to future work.

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

## A DEFINING FEATURE FUNCTIONS

The term "feature" can have different meanings depending on the context, so we clarify our usage here. In this paper, we define a feature as a function over inputs. For example, a binary classification function that returns 1 if an input $x$ is a harmful prompt and 0 if it is harmless constitutes a feature function in our setting.

While binary classification functions are a common example, our definition of a feature function is not limited to them. Any function that clearly describes a property of the input—whether rule-based or expressed in natural language—can be treated as a feature function in our setting. In cases where the function is specified via a natural language description, we can employ an LLM to compute its output.

For the "Object" feature illustrated in Figure 2, we can define it either as "the indirect object name in the prompt" or as "the name in the prompt that appears only once." Both descriptions can be transformed into a concrete algorithm that deterministically assigns a piece of text as a label to any given prompt in the IOI dataset.

# B    APPROXIMATING $P(x; \hat{z})$ VIA ACTIVATION PERTURBATION

In order for the training dataset $\{(x, z)\}$ to faithfully approximate the conditional distribution $P(x; \hat{z})$ as defined, we perturbate the original activations $\hat{z}$ extracted from the target model. Without this perturbation, our experiments show that the conditional generator tends to memorize the exact correspondence between $\hat{z}$ and its original input $\hat{x}$—a behavior we explicitly want to avoid.

If the distance function $d(\cdot, \cdot)$ is a proper metric, i.e., it is symmetric and satisfies $d(z, \hat{z}) = 0 \Rightarrow z = \hat{z}$, then we can inject noise in a way that mirrors the kernel-based conditional distribution $P(x; \hat{z})$. Specifically, we perturb the original activation $\hat{z}$ by sampling a continuous noise vector $r$ from a distribution defined as:

$$p(r) \propto k\left(d(r + \hat{z}, \hat{z})\right),$$

where $k$ is the kernel function used in the definition of $P(x; \hat{z})$.

Let $\tilde{p}(x, z)$ denote the joint density over inputs and perturbed activations, where each input $x$ is paired with a perturbed activation $z = \hat{z} + r$. Then, one can show that evaluating this density at $z = \hat{z}$ recovers the reweighted distribution:

$$\tilde{p}(x, z = \hat{z}) = P(x; \hat{z}).$$

This construction enables us to train the conditional generator on samples of the form $(x, \hat{z} + r)$ such that, at test time, it approximates the desired distribution $P(x; \hat{z})$ when conditioned on the original activation $\hat{z}$.

However, when using measures like cosine distance, as in our experiments, additional complications arise. Specifically, cosine distance satisfies $d(z, \hat{z}) = 0$ for any $z = c\hat{z}$ with $c \geqslant 0$, so the kernel function $k(d(r + \hat{z}, \hat{z}))$ does not induce a proper probability density over the noise variable $r$.

To address this, we introduce a modified distance function:

$$\tilde{d}(z, \hat{z}) = \begin{cases} d(z, \hat{z}), & \text{if } \|\|z\| - \|\hat{z}\|\| < \delta\|\hat{z}\| \\ \infty, & \text{otherwise} \end{cases}$$

This effectively constrains the norm of the perturbed activation to lie within a small band around $\|\hat{z}\|$, ensuring the noise distribution remains well-defined and avoids degenerate directions along the $\hat{z}$ ray.

One can verify that for sufficiently large $\delta$, the equality $\tilde{p}(x, z = \hat{z}) = P(x; \hat{z})$ still holds. However, increasing $\delta$ introduces greater variance into the training labels, making the conditional generator harder to train. In practice, we set $\delta = 0.1$ as a trade-off between theoretical fidelity and empirical stability.

# C    NETWORK AND TRAINING DETAILS

In this section, we describe the general training settings used throughout our experiments.

## C.1    TRAINING OF BACKBONE

As mentioned in the main text, we use GPT-2-small as our backbone model, regardless of the target model we want to interpret. Since GPT-2's tokenizer lacks a predefined begin-of-sentence token, we exploit the original `<|endoftext|>` token to serve as both the begin-of-sentence and end-of-sentence token during backbone fine-tuning.

For the fine-tuning stage, We employ a full-parameter fine-tuning and use the AdamW optimizer with a learning rate of $1 \times 10^{-5}$, while all other hyperparameters are set to their default values in PyTorch. The max token length and the batch size depend on the specific task.

## C.2    TRAINING OF ADDITIONAL LAYERS

For the additional multi-head control layers, we use 32 attention heads, each with a head dimension of 64. The site-specific transformations consist of linear layers with input and output dimensions

that are equal to the target model's hidden dimension. All parameters in the additional layers are initialized using Kaiming initialization, except for the value projection matrices in the control layers, which are initialized to zero. We find that this initialization strategy leads to more stable training dynamics.

In all cases, we use the AdamW optimizer with a learning rate of $1 \times 10^{-5}$, while all other hyper-parameters are set to their default values in PyTorch. A warmup period of 1000 batches is applied, during which the learning rate is linearly increased from zero to $1 \times 10^{-5}$.

For training the backbone and additional layers, we use 4 NVIDIA A800 GPUs. Most training runs complete within 24 hours.

## D    EXPERIMENT SETTINGS AND RESULTS

In this section, we provide a detailed and comprehensive description of the experimental setup for our InverseScope framework. This includes the specific target models and tasks evaluated, the precise methodology used for constructing the training datasets for the InverseScope conditional generator, and the formal definition of the feature functions used in our quantitative analysis.

### D.1    IOI

#### D.1.1    DATASET

To generate IOI inputs, we adopt the templates from the implementation of Conmy et al. (2023). For example, a template such as "Then, [B] and [A] went to the [PLACE]. [B] gave a [OBJECT] to" is instantiated by replacing "[B]" and "[A]" with two random names, while "[PLACE]" and "[OBJECT]" are substituted with random locations and items drawn from predefined sets. This procedure yields approximately 3 million possible combinations. From these, we sample 100,000 examples for training and 5,000 examples for testing. Residual stream activations are collected at the final inference position as activations to be interpreted.

#### D.1.2    FEATURE FUNCTIONS

Given a valid IOI input, we compute the subject and object feature label by checking the frequency of each name in the sentence. Specifically, the name that appears once is assigned as $\text{Object}(x)$, and the name that appears twice is assigned as $\text{Subject}(x)$. This forms a rule-based feature function over input $x$.

#### D.1.3    COMPARING WITH PREVIOUS METHODS

To ensure a fair comparison between our method and InversionView (Huang et al., 2024b), we adopt their experimental setting and use our conditional generator to sample inputs satisfying $d(z(x), \hat{z}) < \epsilon$. However, since the original InversionView method does not incorporate noise during training, it relies on perturbing $\hat{z}$ at sampling time. Accordingly, when sampling with InversionView, we follow their protocol and add Gaussian noise of scale $\epsilon$ to $\hat{z}$ before sending it to the conditional generator.

### D.2    RAVEL

#### D.2.1    DATASET

All sub-datasets of RAVEL are combined into a single corpus, which is used to train both InverseScope and SAE (RAVEL). Following the entity-level train–test split provided in the original dataset, we construct a training set of 100,000 prompts and a test set of 10,000 prompts.

For InverseScope, residual stream activations are collected at the final inference position. For SAE (RAVEL), residual stream activations are collected at all inference positions except the first inference position. Following the setup of Lieberum et al. (2024), for SAE (RAVEL), the activations are shuffled before saving as the training set.

### D.2.2 TRAINING OF SAE (RAVEL)

To ensure a fair comparison, in addition to using the open-sourced SAEs (Lieberum et al., 2024) as baselines, we train a series of JumpReLU SAEs with feature width 4096 specifically on the RAVEL dataset. Training is performed for 100,000 steps with a batch size of 2048 using the AdamW optimizer. We apply an $L_0$ sparsity penalty with regularization parameter $\rho = 5 \times 10^{-5}$. All other hyperparameters follow Lieberum et al. (2024). Due to computation limitations, we only trained SAE (RAVEL) for half the layers (layer 0, 2, 4, ..., 24) in the original model.

### D.2.3 REPRESENTATIVE FEATURE FOR SAE BASELINES

To perform attribute classification using SAE, we assign a representative feature to each attribute in RAVEL. For this, we record which SAE features are activated for 100,000 input activations. For each attribute, we then evaluate all features and select the one with the highest F1 score as the representative feature for that attribute.

### D.2.4 FEATURE/ATTRIBUTE FUNCTIONS

To assign attribute labels to the outputs of InverseScope, we define rule-based feature functions using the templates provided in the RAVEL dataset. Each prompt template in the original dataset is paired with an attribute label—for example, the template "[City] is a city in the country of" is paired with the attribute "City:Country." Accordingly, if the output text of InverseScope matches a template, we assign the corresponding attribute label. Otherwise, we classify it as a null attribute, indicating no match.

This matching criterion is intentionally strict. Nonetheless, InverseScope achieves high labeling accuracy under this rule, as nearly all generated outputs conform to the prompt templates, with very few nonsensical generations observed.

### D.3 ICL

### D.3.1 TASK VECTOR EXPERIMENTS

Task vectors are studied through *activation patching*, a causal intervention technique for probing information encoded in specific activations. Consider the following English-to-French translation setting with two inputs:

- With task examples (few-shot prompt):

$$\text{mile} \rightarrow \text{mile, } \text{cup} \rightarrow \text{coupe, } \text{fact} \rightarrow \text{fait, } \text{lead} \rightarrow$$

- Without task examples (query only):

$$\text{black} \rightarrow$$

In this setup, the model with task examples correctly outputs "plomb" as the French translation of "lead." In contrast, the query-only input is highly likely to yield an unrelated token, since it provides no information about the translation task.

The activation patching procedure proceeds as follows. At a chosen layer, the activation of the first prompt (source) at the final position is recorded and substituted into the forward pass of the second prompt (target) at the same site, while all other activations remain unchanged. If the patched target inference produces "noir," the correct French translation of "black," this indicates that the patched activation encodes task-level information about English-to-French translation. If the output remains unrelated, this suggests no task-level information is encoded. If instead the output is "plomb," the source output token, this indicates that the activation carries output-token information, but it does not reveal whether task-level information is also encoded.

By systematically applying this procedure across layers and positions, it is possible to localize where abstract task representations emerge. As shown in Figure 7, only a few intermediate layers enable correct translation of the second prompt. This observation forms the basis for defining the notion of a "task vector" in prior work.

### D.3.2 DATASET

To generate ICL inputs, we adapt the templates introduced in Hendel et al. (2023). Each prompt is constructed in a 3-shot format:

```
Input: [input_1], Output: [output_1]\n Input:
[input_2], Output: [output_2]\Input: [input_3],
Output: [output_3]\n Input: [input], Output:
```

where each pair ([input$_i$], [output$_i$]) consists of words with equivalent meaning in two different languages.

We focus on six translation tasks: English → French/Italian/Spanish, and their reverse directions, French/Italian/Spanish → English. All six tasks are sampled in equal proportion.

The number of possible prompts is combinatorially large due to the vocabulary size and pairing choices. From this space, we sample 120,000 examples for training and 5,000 distinct examples for evaluation.

### D.3.3 FEATURE FUNCTIONS

The definition of the query feature $\text{Query}(x)$ is straightforward: we identify the word that follows the final "Input:" marker in the prompt. This word serves as the output of the function $\text{Query}(x)$.

To define the task feature $\text{Task}(x)$, we leverage an LLM to assist with labeling. Given an input $x$, we prompt the assistant LLM with a system message asking it to identify the translation task demonstrated in the examples. We use Gemma-2-2B-instruct for this purpose. Additionally, we provide an extra "Mix" task label for cases where the LLM detects more than one type of input–output language pair in the prompt.

## E   MORE EXPERIMENTS

This section provides a series of additional experimental results, comprehensive visualizations, and quantitative analyses designed to substantiate and extend the core findings presented in the main body of the paper. Our primary goal is to provide evidence of the robustness, scalability, and mechanistic precision of the InverseScope framework across various interpretability tasks and model architectures.

### E.1   QUALITATIVE ANALYSIS OF INVERSESCOPE SAMPLED INPUTS

In this subsection, we present a few examples of samples generated by our conditional generator, each conditioned on activations $\hat{z}$ from selected attention heads. Since there are too many attention heads in total, we only visualize results from a few representative sites.

We found L9H3 particularly interesting: as shown in Figure 4 and the examples in Table 1, it does not clearly encode information about the subject or object names. However, one can observe from the sampled examples that it appears to encode the item mentioned in the input $\hat{x}$ —which is "drink" in this case. From a human perspective, such information seems irrelevant to solving the IOI task, yet the model still preserves and transmits it, revealing the complexity of the underlying circuit.

We can also leverage our method for additional forms of analysis. Figure 8 illustrates the relationship between the distance from the original activation and feature consistency. To obtain more diverse input samples $x$, we manually inject additional noise into the activation during sampling. Consequently, the inputs $x$ visualized in this plot do not strictly follow the conditional distribution $P(x; \hat{z})$.

As shown in the figure, attention head L9H6, which exhibits a high feature consistency rate, forms a plateau where inputs with activations satisfying $d(z, \hat{z}) < 0.2$ have a high probability of sharing the same object feature. Similar patterns can be observed for other heads with high feature consistency. These results suggest that further investigation into the structure and distribution of activations $z$ around a given $\hat{z}$ could yield deeper insights into how specific features are encoded and preserved in the model's internal representations.

Table 1: Example of inputs sampled from $P(x; \hat{z})$, where $\hat{z}$ are activations extracted from attention heads L7H9, L9H9, and L9H3 in the IOI task. The activations $\hat{z}$ correspond to the input $\hat{x} =$"After John and Mary went to the store, Mary give a drink to".

(a) L7H9

| $d(z, \hat{z})$ | $x \sim P(x; \hat{z})$ |
|---|---|
| 0.007 | After Mary and Jeffrey went to the garden, Mary gave a drink to |
| 0.011 | After Mary and Michael went to the station, Mary gave a drink to |
| 0.014 | After Mary and Kenneth went to the garden, Mary gave a ring to |
| 0.018 | After Mary and Jeffrey went to the office, Mary gave a computer to |
| 0.017 | After Mary and Jeffrey went to the restaurant, Mary gave a drink to |
| 0.024 | After Mary and Nicole went to the hospital, Mary gave a drink to |
| 0.107 | Afterwards, Mary and Timothy went to the office. Mary gave a drink to |
| 0.196 | Afterwards, Mark and Mary went to the office. Mary gave a drink to |
| 0.183 | Afterwards, Matthew and Mary went to the house. Mary gave a drink to |
| 0.199 | Afterwards, Joseph and Mary went to the garden. Mary gave a computer to |

(b) L9H9

| $d(z, \hat{z})$ | $x \sim P(x; \hat{z})$ |
|---|---|
| 0.031 | Then, Mary and John had a lot of fun at the garden. Mary gave a drink to |
| 0.064 | When Elizabeth and John got a drink at the hospital, Elizabeth decided to give it to |
| 0.066 | When Samuel and John got a bone at the hospital, Samuel decided to give it to |
| 0.073 | After the lunch, Sarah and John went to the house. Sarah gave a kiss to |
| 0.077 | When Erin and John got a computer at the garden, Erin decided to give the computer to |
| 0.088 | After the lunch, Lindsey and John went to the garden. Lindsey gave a drink to |
| 0.092 | After John and Crystal went to the school, Crystal gave a drink to |
| 0.094 | Then, Danielle and John had a lot of fun at the hospital. Danielle gave a drink to |
| 0.163 | After the lunch, Kevin and John went to the hospital. Kevin gave a computer to |
| 0.189 | After John and Steven went to the garden, Steven gave a kiss to |

(c) L9H3

| $d(z, \hat{z})$ | $x \sim P(x; \hat{z})$ |
|---|---|
| 0.028 | After Jacob and Benjamin went to the store, Benjamin gave a drink to |
| 0.033 | Then, Charles and James went to the house. James gave a drink to |
| 0.036 | Then, Mary and Kenneth went to the garden. Mary gave a drink to |
| 0.041 | Then, Charles and James went to the garden. James gave a drink to |
| 0.051 | Then, Jeffrey and James went to the restaurant. James gave a drink to |
| 0.062 | Then, Anthony and Shannon went to the restaurant. Shannon gave a drink to |
| 0.067 | Afterwards, Robert and Jeffrey went to the office. Robert gave a drink to |
| 0.071 | After the lunch, Andrew and James went to the station. Andrew gave a drink to |
| 0.097 | The school James and Jesse went to had a drink. James gave it to |
| 0.137 | Then, Shannon and Kenneth went to the store. Kenneth gave a kiss to |
| 0.143 | The local big house Aaron and Jose went to had a drink. Aaron gave it to |

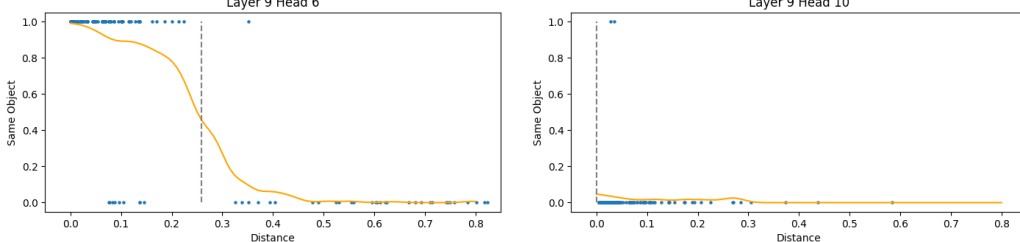

Figure 8: Detailed plot showing the relationship between $d(z, \hat{z})$ and the indicator $\mathbb{I}_{f(x)=f(\hat{x})}$ for the object feature. **Left:** L9H6, **Right:** L9H10. Each blue point represents a sampled pair $(x, z)$. The orange curve shows a kernel-smoothed trend of the sampled points, while the grey line marks the 50% level of the smoothed curve.

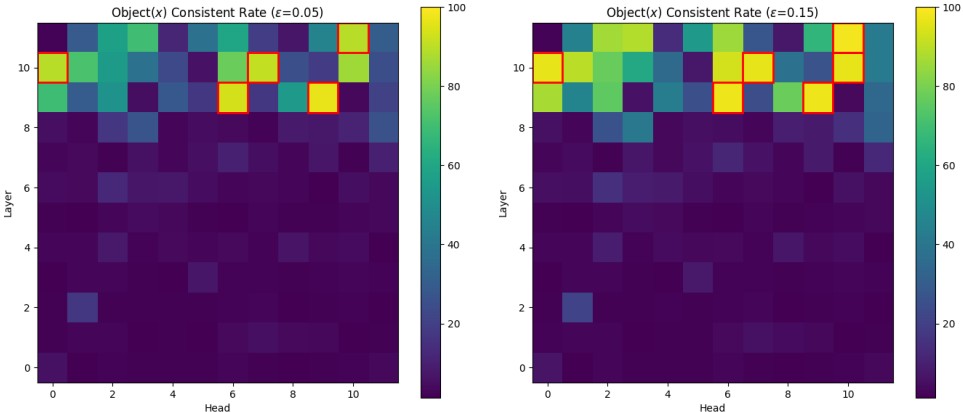

Figure 9: Ablation study of the kernel width $\epsilon$ used during training and sampling. Different choices of $\epsilon$ control the variance of the activation distribution and therefore the variability of sampled reconstructions. **Left:** $\epsilon = 0.05$. **Right:** $\epsilon = 0.15$.

Table 2: Representative layers selected for feature interpretation. We select low, medium and high layers to represent all positions in the target LLM.

| Gemma2 | Representative Layers | Dimension |
|--------|----------------------|-----------|
| 2B | [3, 9, 15, 21] | 2304 |
| 9B | [5, 15, 25, 35] | 3584 |
| 27B | [5, 17, 29, 41] | 4608 |

### E.2 ABLATION STUDY OF THE KERNEL BANDWIDTH $\epsilon$

We conduct an ablation study on the kernel bandwidth $\epsilon$, which controls the neighborhood size used during both training and sampling. Main experimental settings are identical to the IOI experiments in Section 4.1. As shown in Figure 9, varying $\epsilon$ affects the absolute FCR values. A larger $\epsilon$ broadens the neighborhood around $\hat{z}$ and consequently injects more randomness into the reconstructed input distribution. Nevertheless, across all tested settings, the interpretability conclusions remain unchanged: the same attention heads are consistently identified as encoding the Object feature. This robustness indicates that our method is stable under reasonable choices of $\epsilon$, and that the identified feature-bearing heads are not artifacts of a specific hyperparameter setting.

### E.3 DEMONSTRATING SCALABILITY ON LARGE LLMS

In this subsection, we conduct additional experiments to evaluate how InverseScope performs across target models of different sizes. For a controlled comparison, we use the Gemma-2 pretrained model family, selecting the 2B, 9B, and 27B variants. A second factor to control is the activation site under study: larger models differ not only in hidden dimension but also in depth, and therefore offer more potential interpretation sites. To ensure fairness, we select 4 representative layers in each model, corresponding to early, mid, and late stages of computation, and apply InverseScope to residual stream activations at these layers. The selected layers are listed in Table 2.

We follow the same setup as in Section 4.2, training InverseScope on prompts sampled from the RAVEL dataset and the residual stream activations they generated. We evaluate interpretability quality using the attribute classification accuracy computed over test dataset, keeping all evaluation settings identical to the previous experiments.

The results, shown in Figure 10, reveal a clear trend: attribute classification accuracy *improves* as the hidden dimension increases. This behavior runs counter to the common expectation that higher dimensionality should make inversion harder due to the curse of dimensionality. Instead, our findings suggest that larger models form clearer semantically structured representations of attributes in the residual stream. As a result, InverseScope are able to sample inputs that exhibit higher feature consistency rate, leading to higher classification accuracy.

These results demonstrate that InverseScope not only scales to large, production-level LLMs, but also benefits from the higher-quality internal representations present in larger models.

### E.4 VERIFYING ICL MECHANISM IN LLAMA-2-7B

In this subsection, we extend the mechanistic analysis of task vector evolution (as discussed in Section 4.2) to LLaMA-2-7B model. This comparative study demonstrates that the same core mechanism of feature transformation is consistently observed across different large language model (LLM) families.

As shown in Figure 11, LLaMA-2-7B exhibits the same feature consistency rate trend as Gemma2-2B. In layers 12–14 (shaded green), the task feature is clearly detectable, while the query feature is absent—exactly corresponding to the layers where patching produces the correct target output. After layer 15 (shaded red), where the query feature begins to emerge, patching leads to the generation of the source output.

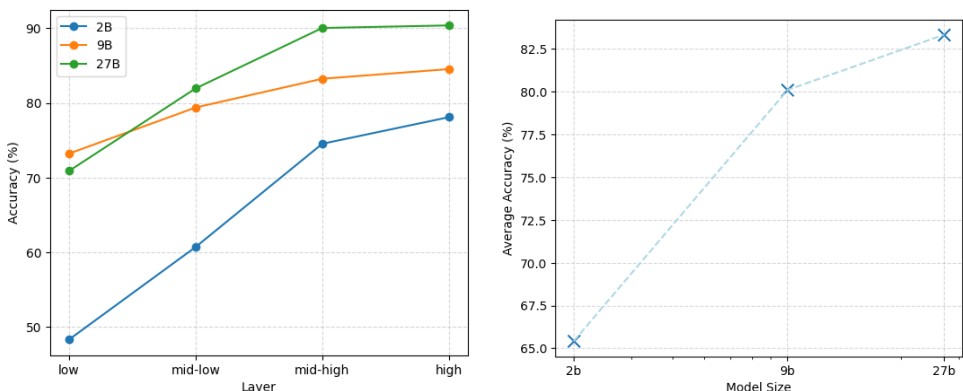

Figure 10: Scaling to higher dimension activations. **Left:** Attribute classification accuracy of different representative layers. **Right:** Average classification accuracy of all representative layers.

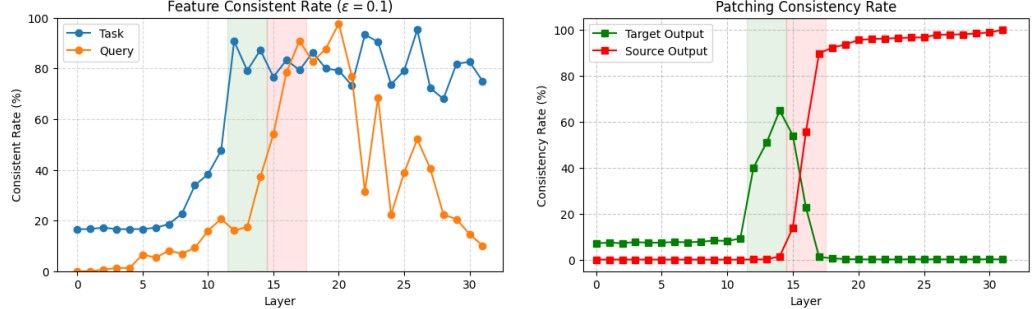

Figure 11: **Left:** Feature consistency rates of the outputs of LLaMA-2-7B's attention layers in the ICL task. Blue represents the task feature. Orange represents the query feature. **Right:** Results of task-vector patching experiments. Green represents the rate at which the patched inference produces the correct target output. Red represents the rate at which it incorrectly produces the source output.

