# OpenReview forum: "InverseScope: Scalable Activation Inversion for Interpreting Large Language Models"
_ICLR.cc/2026/Conference — Submitted to ICLR 2026_

### Official Review · Reviewer_qmH4 · 2025-10-28

**Soundness:** 2
**Presentation:** 3
**Contribution:** 2
**Rating:** 4
**Confidence:** 3

**Summary:**

This paper proposes InverseScope, an activation-conditioned generator designed to interpret neural representations in large language models (LLMs) via activation inversion. Given an activation from a specific site (layer/head), the model defines a conditional distribution over inputs that would yield similar activations and samples from this distribution using a conditional Transformer generator. The authors also introduce the Feature Consistency Rate (FCR), a quantitative metric evaluating whether generated inputs preserve certain features (e.g., subject/object identity).
Experiments show qualitative and quantitative correspondence between activation patterns and encoded semantic features.

**Strengths:**

1. **Clear motivation and formalization:** The paper clearly articulates the challenge of probing activations in LLMs.

2. **Reasonable architectural engineering:** The control-layer conditional generator is a practical and technically sound approach for conditioning a Transformer decoder on internal activations.

**Weaknesses:**

1. **Scalability claim insufficiently supported:** The paper claims to “advance inversion-based interpretability by scaling it to larger open-source LLMs and applying it to practical tasks” (lines 69–70). However, the actual generator is trained on GPT-2 small, and target models are limited to Gemma-2B and LLaMA-2-7B. No experiment demonstrates how the generator behaves with increasing activation dimensionality, even though the authors justify their method by noting that  “the probability that a random input produces an activation close to $\hat z$ decays exponentially with dimensionality” (lines 166–167). If dimensionality scaling is the central motivation, a quantitative study showing how approximation accuracy degrades or stabilizes with dimension is essential.

2. **Limited novelty:** The paper feels incremental in method, with novelty residing mostly in framing rather than technique. The assumption that similar activations imply similar semantics has already been well discussed in prior studies (e.g., Bengio et al., 2013, IEEE TPAMI, as also noted by the authors in lines 87–88). Moreover, the study largely reuses a GPT-2-style conditional language model without structural innovation.  The authors provide no justification for this architectural choice, nor any ablations showing how control-layer design or alternative decoder types affect inversion fidelity.

3. **Shallow and constrained experimental validation:** The experimental validation of InverseScope remains shallow and constrained in scope. All generator experiments rely exclusively on GPT-2-small, with no ablations across different generator sizes or architectures, leaving open questions about model capacity and scaling behavior. Furthermore, the input prompts used throughout the experiments are notably simple and short. Even in the Limitations section (lines  480–481), the authors acknowledge that the current setup does not test complex or compositional language; however, such long, multi-clause prompts would provide a much more meaningful evaluation of scalability and generalization. In addition, while the paper refers to “task-specific input distributions” (lines 241–242), the method is not evaluated on a broader variety of tasks beyond IOI and RAVEL, limiting the evidence for its task-agnostic applicability.

4. **Human-defined feature functions:** Feature functions $f(x)$ are manually constructed by the authors for each task (Appendix D.1.2-lines 691–692, Appendix D.2.4-lines 821–824). However, the paper does not explain why these task-specific, rule-based definitions are the most appropriate way to evaluate feature consistency. If users must manually define $f(x)$ for every new task, the method’s scalability and reproducibility are questionable, since performance could vary substantially depending on how $f(x)$ is specified.

[1] Yoshua Bengio, Aaron Courville, and Pascal Vincent. Representation learning: A review and new perspectives. IEEE transactions on pattern analysis and machine intelligence, 35(8):1798–1828, 2013.

**Questions:**

**Questions for the Authors**

1. Can InverseScope generate diverse and novel sentences (unseen during training) for a given target activation? Since FCR evaluation appears to be closely related to the diversity of generated samples, it would be important to measure and report explicit diversity metrics (e.g., lexical or semantic variance).

2. Could you provide a quantitative analysis showing how approximation accuracy or FCR stability changes as activation dimensionality increases? Does InverseScope maintain inversion fidelity when applied to larger-scale LLMs (e.g., LLaMA-13B or 70B) or to more complex reasoning tasks?

3. The generator architecture is fixed to GPT-2 small. Could you explore alternative generator backbones (e.g., T5, Mistral, Gemma) or larger-scale models? What motivates this specific architectural choice, and would inversion performance or sample diversity change with different configurations?

4. As seen in Figures 2 and 4, only late layers exhibit strong inversion behavior, while early-layer activations appear almost flat. Can the authors provide insight or diagnostic analysis explaining why inversion signals are weak or absent in earlier representations?

**Additional Suggestions**

1. Typographical errors: lines 159–161; Appendix lines 687–688.

2. The related work section is too short to clearly position the paper within recent interpretability research. Expanding it—perhaps in an appendix—would improve clarity and contextual grounding

---

> ### Author Response · Authors · 2025-11-29
>
> We sincerely thank the reviewer their insightful and detailed feedback, which motivated us to conduct extra experiments that makes our work more concrete and rigorous. We apologize for the delay in this response, which was necessary as we conducted additional, computationally intensive experiments to address the critical points raised, particularly concerning scalability and diversity.
>
> > Can InverseScope generate diverse and novel sentences (unseen during training) for a given target activation? Since FCR evaluation appears to be closely related to the diversity of generated samples, it would be important to measure and report explicit diversity metrics (e.g., lexical or semantic variance).
>
> We appreciate this excellent suggestion regarding sample diversity. Yes, InverseScope generates diverse and novel sentences. To quantitatively demonstrate this, we calculated the Type-Token Ratio (TTR) for the generated inputs in the RAVEL task, comparing them against samples randomly drawn from the original dataset and repeated sampling the input prompt. We sampled $N=16$ reconstructions per prompt for the comparison.
>
> The results, presented below, show that the Mean TTR of our reconstructed inputs is statistically close to that of the random samples from the original dataset. This confirms that InverseScope does not simply repeat common phrases but successfully generates novel, diverse inputs consistent with the target activation, effectively capturing the input distribution conditioned on the activation.
>
>
> | Method | Mean TTR (Std) | Types/Tokens |
> | :--- | :--- | :--- |
> | Reconstruct | 0.5312 (0.2019) | 68.1/132.9 |
> | Random | 0.5984 (0.0626) | 83.0/139.8 |
> | Repeat | 0.0506 (0.0094) | 6.6/137.1 |
>
>
> > Could you provide a quantitative analysis showing how approximation accuracy or FCR stability changes as activation dimensionality increases? Does InverseScope maintain inversion fidelity when applied to larger-scale LLMs (e.g., LLaMA-13B or 70B) or to more complex reasoning tasks?
>
> This is a crucial point, and we agree that a quantitative analysis of scaling with dimensionality is essential. This is precisely why we delayed our response: we performed a set of computationally intensive experiments by applying InverseScope to larger target models with higher activation dimensionalities: Gemma2-9B ($d=3584$) and Gemma2-27B ($d=4608$) on the RAVEL dataset.
>
> Contrary to the normal expectation that inversion fidelity decays exponentially with dimension, we observed that the RAVEL prediction accuracy actually increased as the activation dimension increased. This result is highly encouraging and, upon reflection, reasonable: a higher-dimensional hidden state in a larger, more capable target model typically contains a richer, more comprehensive encoding of the input, enabling InverseScope to recover more high-fidelity information. This result strongly supports our claim that InverseScope successfully scales to target models (up to 27B). We have included a detailed discussion and full results in Appendix E.3 of the revised manuscript.
>
> > The generator architecture is fixed to GPT-2 small. Could you explore alternative generator backbones (e.g., T5, Mistral, Gemma) or larger-scale models? What motivates this specific architectural choice, and would inversion performance or sample diversity change with different configurations?
>
> The choice of GPT-2 small as the generator backbone was primarily motivated by computational cost and efficiency. Since our method is designed to condition the generator on a relatively small, task-specific input distribution (as opposed to a general corpus), the capacity of GPT-2 small was empirically determined to be sufficient for effectively modeling such distributions. While we agree that a larger or alternative architecture (like Mistral or Gemma) could potentially improve inversion fidelity, given the strong performance demonstrated by GPT-2 small in our current constrained settings, we estimate that the performance gain would be marginal relative to the substantial increase in training and inference cost. The key novelty remains the conditional mechanism layered onto the backbone, not the specific base model.

---

> ### Author Response · Authors · 2025-11-29
>
> > As seen in Figures 2 and 4, only late layers exhibit strong inversion behavior, while early-layer activations appear almost flat. Can the authors provide insight or diagnostic analysis explaining why inversion signals are weak or absent in earlier representations?
>
> We believe the reason for this observation lies in the nature of the specific features we are targeting. The object name feature, which is crucial for the final output prediction, is naturally encoded in the deeper, later layers of the model, consistent with the expected function of high-level semantic processing. The subject name feature, in contrast, exhibits stronger concentration in the middle layers. If we were to focus on more local, token-level features, these would be expected to show stronger inversion signals in the shallower, earlier layers.
>
> > Limited novelty. Moreover, the study largely reuses a GPT-2-style conditional language model without structural innovation.
>
> We respectfully clarify the novelty: we agree that the backbone in our method is a standard decoder Transformer architecture. However, our core technical contribution and novelty reside in the control layers. This specific architectural modification allows the standard decoder to condition not just on a prefix token sequence, but critically, on a high-dimensional continuous activation vector. This design is robust and practical for scaling inversion-based methods to target LLMs with diverse internal geometries, which is the main claim of the paper.
>
> > Shallow and constrained experimental validation. ... leaving open questions about model capacity and scaling behavior.
>
> We appreciate the feedback on the scope of our validation. We feel there might be a slight misunderstanding regarding the nature of our "scaling" claim: we primarily aim to scale inversion-based interpretability to much larger target LLMs, not necessarily scaling the InverseScope generator itself (though the latter is a valuable future direction). Our new experiments on Gemma2-27B directly address the robustness of our method when applied to large target models. Regarding task variance: we acknowledge that broader validation is ideal. However, we would like to point out that our current validation includes 7 distinct tasks in total: the IOI task, the ICL translation task and the RAVEL dataset, which itself is composed of 5 distinct tasks. This provides a non-trivial level of evaluation across different types of language features.
>
> > Typographical errors.
>
> We thank the reviewer for pointing out the typographical errors. These have been corrected in the revised manuscript.
>
> > The related work section is too short to clearly position the paper within recent interpretability research. Expanding it—perhaps in an appendix—would improve clarity and contextual grounding.
>
> We agree that clearly positioning our work within the broader interpretability literature is important. While we have kept the main paper concise due to page constraints, we have improved the related works section to better illustrate the relationship between our work and existing works.

---

### Official Review · Reviewer_FqeL · 2025-10-31

**Soundness:** 2
**Presentation:** 3
**Contribution:** 2
**Rating:** 6
**Confidence:** 2

**Summary:**

The authors present a new technique to map from activations to plausible input sequences that would produce similar activations. To do this, the authors train a conditional generation model. This model is a small transformer model trained on next token loss but conditioned on the latent activations of the model we would like to interpret. Instead of training a site specific model, which would be computationally expensive, they train translator linear layers for each site, training a single unified conditional model. Their conditional model is a finetuned of GPT-2 small, but they apply it to larger models.

**Strengths:**

The proposed method is more sample efficient than other inversion methods.

The proposed method correctly identifies some of the attention heads that are important for the IOI circuit in GPT2.

The accuracy on the classification task on the RAVEL benchmark surpasses that of SAEs.

**Weaknesses:**

The 'inverting' model has to be re-trained for each specific task.

Although this can be said of several interpretability methods, it is not clear here if the latent representations have any causal link to the mechanisms of the underlying model and it is not obvious how to test the predictions made.

On the IOI task, the 'ground' truth heads were correctly identified by the consistency rate, but there were also other several heads that had similar consistency. In a world where 'ground' truth labels don't exist, it is not obvious how much this method could be used to identify relevant components.

**Questions:**

I think this sentence (lines 136-137) is not well constructed `Given the distribution P(x; ˆz), we propose a three-step pipeline for feature interpret, involving hypothesize, formalize and evaluate:`

I couldn't quite understand how many samples were used to finetuned the InverseScope model for each task.

Could an LLM distinguish between the 'true' input and the generated examples?

---

> ### Author Response · Authors · 2025-11-25
>
> We thank the reviewer for their valuable feedback and thoughtful assessment of our work. We address the raised weaknesses and questions below.
>
> ### Responses to questions
> > I think this sentence (lines 136-137) is not well constructed "Given the distribution P(x; ˆz), we propose a three-step pipeline for feature interpret, involving hypothesize, formalize and evaluate:"
>
> We agree that the original sentence is awkwardly constructed and appreciate the reviewer bringing this to our attention. We will revise the text for improved clarity and flow. The corrected version will read:
>
> "We propose a three-step interpretation pipeline based on the defined distribution $P(x; \hat{z})$, which consists of hypothesis generation, formalization, and evaluation."
>
> > How many samples were used to finetuned the InverseScope model for each task?
>
> We clarify that the InverseScope model was trained using a specific number of samples for each task, with the same dataset being used for both the backbone fine-tuning of the target model and the subsequent InverseScope training. The exact counts are detailed in Appendix D of the submission: we utilized 100,000 samples for both the IOI and RAVEL, and 120,000 samples for the ICL.
>
> > Could an LLM distinguish between the 'true' input and the generated examples?
>
> Ideally, an LLM should not distinguish between the "true" input and a generated example. Our goal is to produce plausible alternative inputs (fluent natural language, as shown in Appendix E.1 of the revised paper) that generates similar activations, not an identical reconstruction. Therefore, both the original and the generated texts should be recognized by a sophisticated language model as valid prompts within the relevant data distribution.
>
> ### Responses for weakness
> > The link between latent representations and model mechanisms
>
> We acknowledge the reviewer’s concern regarding the causal link between latent representations and underlying model mechanisms. Our long-term objective is indeed to understand model mechanisms, and we view the interpretation of latent representations as a necessary intermediate step toward this goal. While our method is correlational in nature, it provides structured hypotheses about what an activation encodes, which can then inform and prioritize targeted causal interventions. In this sense, analyzing latent representations serves as a foundation for subsequent mechanistic studies, rather than a substitute for them.
>
> > Other heads in IOI task that has high FCR
>
> We agree that Figure 4 shows multiple heads with high FCR scores in addition to the primary Name Mover Heads. However, this observation aligns with prior findings in the original IOI paper, which identified not only Name Mover Heads but also Backup Name Mover Heads that play supporting roles in the same circuit. The additional heads with high FCR in our results also coincide with these Backup Heads. Therefore, we interpret this as reflecting genuine structural redundancy in the IOI circuit and the model’s internal organization, rather than noise introduced by our method.

---

> > ### Comment · Reviewer_FqeL · 2025-11-27
> >
> > I thank the authors for replying to my review. I will wait for all the other reviews to be replied to before deciding wether to increase my score.

---

### Official Review · Reviewer_SbTY · 2025-11-02

**Soundness:** 3
**Presentation:** 3
**Contribution:** 3
**Rating:** 8
**Confidence:** 3

**Summary:**

The authors propose InverseScope, a framework for interpreting neural activations by reconstructing the textual inputs that triggered them. In order to do so, the authors propose a conditional generation architecture, where a projection of the representation from the original network is used to conditionally influence reconstruction of the original network’s input. The reconstruction network is initialized to GPT-2, and then its parameters are frozen and only the conditional set is trained to induce the reconstructon behavior after a solid language modeling base.
The authors perform experiments on the IOI task and RAVEL datasets, showing that the proposed architecture improves attribute identification over SAE-based baselines in the latter. In a follow up analysis, the authors also use their framework to analyze where task-specific information obtained from ICL is encoded in the model, and confirm findings from previous works which suggested that these features are encoded in the middle layers.

Overall the paper is well written and easy to follow. The work opens up interesting avenues for inspecting knowledge encoded within LLM latent representations.

**Strengths:**

- Proposes a novel framework for interpreting LLM internals
- Operationalizes the framework with a conditional generation architecture
- Results on IOI and RAVEL show the method is promising and more accurate compared to SAE-based alternatives
- Interesting analysis sheds light where task-specific features from ICL are encoded within the model

**Weaknesses:**

- It would be interesting to also show qualitative samples of reconstructed inputs, as well as failure cases.
- I think it is too strong to consider a conditional LM as a standalone contribution, as such architectures have been widely used prior.
- The related works section feels quite thin. Namely, conditional generation based on model latents, and the connections to the perspective of variational autoencoders seem relevant - but this is quite minor.

**Questions:**

See above

---

> ### Author Response · Authors · 2025-11-25
>
> We sincerely thank the reviewer for their positive assessment. We have addressed the questions and minor concerns raised in the review as follows:
>
> > Qualitative samples of reconstructed inputs and failure cases
>
> We provide several examples in Appendix E.1 for reconstructed inputs of the IOI task. Reconstructed inputs for RAVEL and ICL follow a similar pattern.
>
> Regarding failure cases, our method does not produce nonsensical sentences, and all generated samples are fluent natural language. In this sense, there are no failures from a purely semantic perspective. However, in some cases, the generated inputs do yield activations that are distant from the given target activation. We suspect this may be due to limited training data, although we do not currently have conclusive evidence to support this explanation.
>
> > Too strong to consider a conditional LM as a standalone contribution. & The related works section.
>
> We thank the reviewer for pointing this out and agree that the current framing may overstate the novelty of the conditional LM architecture itself. We will revise the paper to clarify that our contribution does not lie in proposing a new conditional generation architecture, but rather in its specific application to activation inversion for interpretability, as well as the associated design choices and empirical efficiencies in this context.
>
> We also acknowledge that the Related Work section is currently insufficient. In the revised version, we will expand it to include prior work on conditional generation from latent representations and discuss connections to variational autoencoders and related frameworks. We will also discuss the parallel inspiration drawn from conditioning mechanisms found in models like ControlNet in the diffusion literature, and better situate our method within this broader literature.

---

### Official Review · Reviewer_68Gf · 2025-11-03

**Soundness:** 2
**Presentation:** 2
**Contribution:** 2
**Rating:** 4
**Confidence:** 4

**Summary:**

The paper is motivated by the goal of finding "assumption-free" interpretability methods, that don't rely on the linearity and/or sparsity assumptions present in many contemporary tools, such as sparse autoencoders (SAEs). As such, there are no assumptions made on compositionality, and the paper exclusively focuses on the "local" interpretability problem of creating a general tool for interpreting individual LLM activations, as opposed to the "global" problem of discovering compositional structure within the entire space of activations.

The intuition for the method is as follows: suppose we invert the activation and get back a text input having some salient property $p$. We might think that the activation encodes that "the input has property $p$". We can check this by sampling many activations "close" to it in the embedding space, invert them, and check if they have the same property. If they all do, we  have some reason to believe the activation encodes $p$; if it's a random mix of have / not have $p$, we have some reason to think maybe this activation is not sensitive to $p$.

To summarize the presentation of the methodology from Section 2, the paper proposes a method to interpret activations $z$ in LLMs by:
1. approximately inverting noisy versions of $z$ back to input texts;
2. using these texts to formulate a hypothesis for a feature $f$ of text that could be represented by $z$;
3. checking the hypothesis by evaluating the probability (over a task-related text distribution $x\sim D$) that activations close to $z(x)$ have the same value of $f$ as $x$. Notably, there is no condition that activations far from $z(x)$ should have a *different* value of $f$ from $x$; such activations simply don't matter for the objective by which the hypothesis is evaluated.


To approximately invert activations, the authors train a language model conditioned on internal representations, trained to generate input texts that result in representations close to the wanted internal representation. This is effectively a next-token prediction objective.

Note that this method provides a way to classify activations $z$ whenever we have some function $f$ from texts to a finite set of classes. This is because we can approximately invert $z$ and evaluate $f$ on the inverted text. In that way, the method is similar to the well-known linear probe method, but without the linearity assumption.

Experiments include:
- identifying which heads in GPT-2 represent which features in the IOI task, a well-studied simple language task whose circuit has been mapped extensively in prior work.
- benchmarking against SAEs on the RAVEL dataset, where InverseScope is compared to using individual SAE features as classifiers.
- studying the layers in which task vectors emerge

**Strengths:**

The paper tackles a somewhat under-investigated question in the interpretability literature overall: can we have an "oracle" that simply tells us what features of the input text a given activation "cares about", without relying on assumptions like linearity or sparse coding? The work makes the assumption that activation semantics is "continuous", which seems reasonable as far as assumptions go.

The writing is clear & easy to follow.

**Weaknesses:**

- The contribution over the prior work [1] is relatively incremental. The prior work also trains an activation inverter. The main contribution of the current work is not in methodology, but in the kernel used for approximate inversion and in the experiments.
- The approximate activation inversion process complicates and obfuscates the method, as it introduces hyperparameters (the noise scale and the "width" of the kernel) with an unknown role in the final results. Additionally, the inversion only works on a limited task dataset, limiting the overall applicability of the method. This means that the method can only generate hypotheses for concepts that exhibit variation in the task dataset, unlike an SAE for example, which can generate hypotheses based on concepts in the entire pre-training distribution. In other words, the question being answered here is not "What is the model thinking about when processing the input that created this activation" but "What *dimensions of variation in the task dataset* is the model thinking about when processing the input", which is subtly but crucially different.
- This is at heart a correlational method (no causal experiments are performed), and as such there's only limited interpretability utility to be found in it. I won't make this objection in detail, as it is analogous to the challenges to probes as an interpretability tool that have already been raised in the literature. See work by Belinkov and colleagues, e.g. “Probing Classifiers: Promises, Shortcomings, and Advances” or “Probing the Probing Paradigm: Does Probing Accuracy Entail Task Relevance?”
	- Related to that, the authors say "Crucially, while the original benchmark evaluates interpretability via causal interventions on model behavior, we instead focus on a more fundamental question: assessing the method’s fidelity in identifying the correct attribute encoded within the activation itself" (line 377) - I disagree that this is more fundamental.
- As the authors readily point out, in general there's no obvious way to generate feature hypotheses for step 2. of the method.



[1] Xinting Huang, Madhur Panwar, Navin Goyal, and Michael Hahn. Inversionview: A
general-purpose method for reading information from neural activations

**Questions:**

What is the sensitivity of the results to the hyperparameters involved in the inversion method?

In general in interpretability, we always know the input an activation came from. Since in your method you don't have a general method to generate the hypothesis $f$, a plausible alternative is to skip the activation inversion altogether, and instead formulate a hypothesis by perturbing the input in salient ways and measuring activation distance. How do you think about the tradeoffs here?

---

> ### Author Response · Authors · 2025-11-25
>
> We thank the reviewer for their careful reading and constructive feedback. Below we address the main concerns and questions.
>
> ### Responses to Questions
>
> > What is the sensitivity of the results to the hyperparameters involved in the inversion method?
>
> Our results are empirically robust to the choice of inversion hyperparameters. While changing the kernel width $\epsilon$ does affect the absolute value of the FCR, the qualitative interpretability conclusions remain stable. For example, variations in $\epsilon$ do not alter which attention heads are identified as Name Mover Heads in IOI, nor do they change the identified layers responsible for task vector formation. (We have provided extra results in Appendix E.2 of the revised PDF). This indicates that the method’s interpretive outcomes are not driven by fragile hyperparameter tuning but instead reflect stable properties of the underlying activations.
>
> > A plausible alternative is to skip inversion and instead perturb the input in salient ways and measure activation distance.
>
> Direct input perturbation faces a fundamental combinatorial challenge: the space of meaningful perturbations grows exponentially with sequence length and linguistic complexity. While human or LLM-guided perturbations could reduce this space, they inevitably introduce uncontrolled biases whose influence is difficult to quantify. In contrast, activation inversion provides a principled, model-grounded way to explore the local neighborhood of an activation, allowing hypotheses to emerge directly from the model’s internal geometry.
>
> ### Responses to Weaknesses
>
> > Incremental contribution over InversionView [1]
>
> We agree that both works share the overarching goal of activation inversion. However, our contribution is not limited to introducing a different kernel. We propose a new architectural design for the inversion model that significantly improves efficiency and stability. As demonstrated in Section 4.1, our architecture achieves higher sample efficiency compared to [1]. This improvement is critical for scaling inversion-based interpretability to larger models and higher-dimensional activation spaces, which is a central bottleneck in current practice.
>
> > Correlational method or causal method
>
> We fully acknowledge the importance of causal analysis for drawing concrete interpretability conclusions. However, correlational and causal methods are not mutually exclusive. Rather, they play complementary roles. Correlational approaches are more tractable and scalable, thus acting as foundational tools to guide subsequent causal experiments—similar to how SAE bases support circuit studies.
>
> > About task specific limitation
>
> We believe this observation reflects a current computational constraint in our experiments, rather than a fundamental limitation of the InverseScope method. We applied the method to task-specific distributions primarily to manage the high computational cost associated with sampling and training over the entire pre-training distribution of a large LLM. Crucially, we demonstrated the method's potential for generalization in Section 4.2 by successfully applying it to the RAVEL dataset, which incorporates five diverse tasks. The effectiveness of InverseScope in identifying salient features across these varied tasks provides evidence of its ability to capture generalizable representations, and scaling our approach to a larger, task-agnostic corpus is a direct priority for future work.

---

### Author Response · Authors · 2025-11-29

We sincerely appreciate the reviewers' insightful comments and thoughtful questions, which have driven us to conduct more comprehensive experiments to substantiate the claims and conclusions presented in our paper. We have uploaded a new version of the manuscript that incorporates these improvements.

The main alterations in the revised paper are as follows:

**1. New Ablation Studies and Scalability Experiments (Appendix E)**

We have added a new, comprehensive Appendix E "More Experiments" to rigorously address empirical concerns and demonstrate the robustness of our framework.

- **Hyperparameter Ablation:** We include an ablation study on the critical hyperparameter, the bandwidth $\epsilon$, demonstrating that our InverseScope method is robust and its performance remains stable across a sensible range of bandwidth choices.
- **Scalability to Larger LLMs:** We present new experimental results demonstrating the applicability and efficacy of InverseScope on significantly larger models: Gemma2-9B and Gemma2-27B. These results definitively showcase the method's ability to scale effectively for interpreting large LLMs.

**2. Clarified and Improved Related Work**

As suggested by some reviewers, we have reorganized and refined the Related Work section. The section now provides a much clearer explanation of how InverseScope relates to existing literature, specifically by contextualizing our approach within prior work in conditional generation and other contemporary mechanistic interpretability methods. This change helps better position our work among the existing works.

We have also taken care to fix minor typos and resolve typesetting issues throughout the revised version of the paper.

We are again thankful for the detailed feedback provided by the reviewers. Their critique has significantly helped us enhance the quality and rigor of our paper.

---

### Meta-Review · Area_Chair_53YL · 2026-01-11

**Summary:**

Few main concerns:

* The contribution seems incremental compared to [1].
* The approximate activation inversion process complicates and obfuscates the method
* Generality issue (This means that the method can only generate hypotheses for concepts that exhibit variation in the task dataset, unlike an SAE for example, which can generate hypotheses based on concepts in the entire pre-training distribution.)
*  the inversion only works on a limited task dataset, limiting the overall applicability of the method.
* it is not clear here if the latent representations have any causal link to the mechanisms of the underlying model
* task-specific limitation
* Shallow and constrained experimental validation

**Reviewer Concerns:**

Authors did respond to lots of concerns by different reviewers, however, I believe a few major ones (novelty, task-specific limitations, and generality of the approach) are still unanswered convincingly.

**Reviewer Scores:**

I don't think that the rebuttal (in its original format) would have changed the scores to a point where it'd be a clear accept.

---

### Decision · Program_Chairs · 2026-01-26

Reject